# Enhancing emotion regulation with an in situ socially assistive robot among LGBTQ+ youth with self-harm ideation: protocol for a randomised controlled trial

A Jess Williams [1,2] Seonaid Cleare,[1,3] Rohan Borschmann,[4,5] Christopher R Tench ,[6] James Gross,[7] Chris Hollis,[8] Amelia Chapman-Nisar,[9] Nkem Naeche,[9] Ellen Townsend ,[10] Petr Slovak,[1,5] On behalf of Digital Youth

For numbered affiliations see end of article.

**Correspondence to**
Dr Seonaid Cleare;
seonaid.cleare@glasgow.ac.uk

## ABSTRACT

**Introduction** Purrble, a socially assistive robot, was codesigned with children to support in situ emotion regulation. Preliminary evidence has found that LGBTQ+ youth are receptive to Purrble and find it to be an acceptable intervention to assist with emotion dysregulation and their experiences of self-harm. The present study is designed to evaluate the impact of access to Purrble among LGBTQ+ youth who have self-harmful thoughts, when compared with waitlist controls.

**Methods and analysis** The study is a single-blind, randomised control trial comparing access to the Purrble robot with waitlist control. A total of 168 LGBTQ+ youth aged 16–25 years with current self-harmful ideation will be recruited, all based within the UK. The primary outcome is emotion dysregulation (Difficulties with Emotion Regulation Scale-8) measured weekly across a 13-week period, including three pre-deployment timepoints. Secondary outcomes include self-harm (Self-Harm Questionnaire), anxiety (Generalised Anxiety Disorder-7) and depression (Patient Health Questionnaire-9). We will conduct analyses using linear mixed models to assess primary and secondary hypotheses. Intervention participants will have unlimited access to Purrble over the deployment period, which can be used as much or as little as they like. After all assessments, control participants will receive their Purrble, with all participants keeping the robot after the end of the study. After the study has ended, a subset of participants will be invited to participate in semistructured interviews to explore engagement and appropriation of Purrble, considering the young people's own views of Purrble as an intervention device.

**Ethics and dissemination** Ethical approval was received from King's College London (RESCM-22/23-34570). Findings will be disseminated in peer review open access journals and at academic conferences.

**Trial registration number** NCT06025942.

## INTRODUCTION

Self-harm, defined as the intentional poisoning or injury of self, irrespective of intention,[1] is a key health concern among sexual orientation and/or gender identity minorities, LGBTQ+ populations.[2]

## STRENGTHS AND LIMITATIONS OF THIS STUDY

⇒ This is the first randomised controlled trial to explore the impact of access of Purrble, a socially assistive robot, compared with waitlist control on emotion regulation (ER) difficulties within LGBTQ+ youth who have current experiences of self-harmful ideation.

⇒ Purrble was codesigned with youth to support ER in situ.

⇒ The study was codesigned with young people who have experience of mental health difficulties (Sprouting Minds), including a detailed safeguarding procedure.

⇒ Participants will not be blinded to participant group due to the nature of the intervention.

Internationally, young LGBTQ+ people report higher prevalence of self-harmful thoughts and behaviours, anxiety, depression and substance misuse when compared with their cisgender, heterosexual peers.[3–8] It is well documented that those with a history of self-harm are at greater risk of suicide,[9] and recent evidence indicates that LGBTQ+ youth are 3–6 times more likely attempt suicide than their cisgender, heterosexual counterparts.[10 11] Despite the considerable risk of self-harm and associated adverse outcomes,[12 13] there is a lack of evidence-based interventions to support LGBTQ+ youth struggling with self-harmful thoughts and behaviours.

Youth who self-harm often do not seek professional help[14 15] and those that do find services (eg, Child and Adolescent Mental Health Services, Accident and Emergency or social services) to be less helpful sources of support[16] or rarely reattend services.[17] Factors relating to this can include negative attitudes and behaviours/treatment from healthcare

staff (eg, withdrawing pain-reducing medication for wound treatment[18]), concerns about confidentiality[19] or perceived stigma surrounding self-harm.[20 21] Among LGBTQ+ people, help-seeking is even more complex, with one in seven avoiding services due to fears of discrimination on the basis of their sexual orientation or gender identity.[22] Therefore, community-based interventions may be more appropriate to support LGBTQ+ youth engaging in self-harm.

As LGBTQ+ youth are frequent users of digital technologies,[23–26] there is an opportunity for digital interventions to support those struggling with self-harm and other mental health difficulties. Evidence suggests that digital interventions support youth to bypass various barriers to help-seeking, such as lack of accessibility, anticipated stigma, inadequate resources and the desire to be self-reliant,[27] which are compounded by unique challenges facing LGBTQ+ youth (eg, concerns about experiencing stigma or discrimination as a minority[28]).

At present, the field concerning digital interventions among LGBTQ+ youth is small, yet those available are perceived as feasible, acceptable and relatively effective.[29] However, most focus on physical health such as risk reduction or management of sexually transmitted illnesses (STIs[29]), with few concerning mental well-being.[30–33] These mental health interventions are typically perceived positively by LGBTQ+ youth,[30 32–34] with mixed findings reported by the three studies which considered the impact of the intervention on participants; (1) Rainbow SPARX[32]; (2) an online writing intervention[31]; and (3) QueerViBE.[30] In their pilot trial, Rainbow SPARX (a didactic PC game using cognitive behaviour therapy principles) was associated with large reduction for depressive (d=1.01) and anxious (d=0.95) symptoms.[32] QueerViBE (a series of brief, interactive videos designed for transgender and gender diverse youth) found a moderate decrease in psychological distress (d=0.63) when compared with the control group.[30] However, the expressive writing intervention demonstrated no difference in depressive symptoms in their randomised controlled trial.[31] Therefore, while limited, digital interventions are feasible, acceptable, and potentially effective for improving mental health among LGBTQ+ youth. However, there are currently no evidence-based digital interventions targeting LGBTQ+ youth who struggle with self-harm.

While self-harm among LGBTQ+ youth can be associated with multiple risks, complex experiences and unique stressors,[35–37] a common issue is often emotional dysregulation.[38–41] Experiencing difficulties with emotion regulation (ER) is a well-known transdiagnostic risk factor,[42–44] which can be associated with higher risk of self-harm across ages, settings and genders.[45 46] Typically, LGBTQ+ populations report greater difficulties with ER,[40 41] which explains in part the association between LGBTQ+ identity and self-harm.[40 41] Examining how ER can be better supported in young LGBTQ+ people who self-harm through digital intervention may be a helpful preventative strategy to aid LGBTQ+ youth broadly.

To address this, a pilot study was conducted with LGBTQ+ youth who had recent experiences of self-harmful thoughts and/or behaviours using an in situ, ER intervention device, Purrble,[47] designed to provide in-the-moment support, see Intervention section for further details. Purrble was originally developed for children in moments of situated distress, but it has since been well-accepted across child and student populations delivering notable benefits for ER.[47–50] Among a small sample of 21 LGBTQ+ young people, Purrble was found to be a feasible and acceptable intervention with continued device engagement across a 2-week deployment.[47] Notably, access to Purrble was also associated with a reduction in anxiety symptoms and self-harmful thoughts. Qualitative data indicated that this was linked to Purrble supporting ER practices (eg, grounding, soothing) to prevent young people acting on their self-harmful urges and, in some cases, preventing them from considering self-harm at all.[47] This is the only study to date which has explored the impact of a socially assistive robot (SAR) among LGBTQ+ youth, who are at risk of self-harm.[47] Based on these findings and Purrble's original design to support in situ, bottom-up ER,[48 49] it appears that mental health outcomes such as anxiety and self-harm[47 50] are guided by the proximal change in ER.

SARs have previously been used to support children in education,[51] family[49] or health settings,[52 53] as well as adults with health conditions such as dementia or physical illnesses.[54 55] These studies have shown promising results in the context of motivation, skill development and enhancement, as well as supporting mental health outcomes, for example, reducing loneliness and stress.[49–55] Similarly, students and at-risk young people have described Purrble robots as a mechanism for comfort and distress relief.[47 50] However, an ethical challenge raised in SARs literature is the use of these device as a replacement for humans, which could incur negative impacts considering social isolation.[56] Therefore, research using SARs should be mindful of this, considering this influence in process analysis, and have additional procedures to prevent over-reliance on these devices.

Although, early data relating to Purrble robots is promising,[47] there is a lack of robust quantitative data on the impact of the Purrble robot in a wider sample of LGBTQ+ young people who have self-harmed. Evidence is therefore urgently needed to evaluate the efficacy of Purrble in (a) delivering measurable changes in ER when compared with a control group and (b) the extent to which this impacts the frequency of self-harmful thoughts and/or anxiety symptoms.

## STUDY OBJECTIVES
### Primary objective
The primary objective of this study is to evaluate the impact of having access to the Purrble robot, compared with a waitlist control, on ER difficulties (Difficulties with

**Table 1** Overview of assessment design for both participant groups

| Surveys | Pre deployment (weeks 1–3) | | | Deployment (weeks 4–13) | | | | Follow-up (week 13+) |
|---|---|---|---|---|---|---|---|---|
| | T(−2) | T(−1) | T(0) | T1–T4 | T5 | T6–T9 | T10 | |
| Register interest+screening | X | | | | | | | |
| Consent | X | | | | | | | |
| Main assessment | X | X | | X | | X | | |
| Extended assessment | | | X | | X | | X | |
| Qualitative interviews | | | | | | | | X |

Emotion Regulation Scale-8 (DERS8)) among LGBTQ+ young people with self-harmful thoughts.

### Secondary objectives

The secondary objectives are (1) to investigate the impact of having access to Purrble on changes to LGBTQ+ young people's self-harmful thoughts over the trial period, in comparison to a waitlist control group and (2) to investigate the impact of Purrble on changes in symptoms of anxiety (Generalised Anxiety Disorder Questionnaire-7 (GAD-7)) and symptoms of depression (Patient Health Questionnaire (PHQ-9)) over the trial period, in comparison to waitlist controls. Finally, this will be the first opportunity to assess whether Purrble remains appealing and helpful to LGBTQ+ youth over an extended period.

### METHODS
### Trial design

The study is a two-arm randomised controlled trial comparing an intervention group (Purrble robot) with a waitlist control group. The trial period is across 13 weeks, built of 3 pre-deployment assessments and 10 deployment assessments, using weekly, self-reported, validated surveys hosted by Qualtrics (see table 1). The intervention period will commence once Purrble has been deployed to the intervention group, week 4 (T1).

Analyses will be conducted and reported in accordance with the Consolidated Standards of Reporting Trials ([57][58]), with consideration given to the recommendation of psychological interventions.[52] Outcomes will be assessed 13 times across a 13-week period, including 3 baseline assessments and 10 weeks of deployment, with Purrble being delivered in time for week 4 (T1).

### Intervention

The intervention takes the form of an interactive plush toy-robot (figure 1), which was codesigned with children to support in-the-moment soothing.[48][49] Purrble is framed as an anxious creature, in need of care and attention when it feels distressed. Embedded electronics are used to produce vibration patterns simulating heartbeats such as 1) frantic and anxious, and (2) slow, steady and relaxed. When held, the device emits a frantic heartbeat which can be slowed by stroking movements registered by embedded sensors. Once the device has been 'soothed' for long

enough, the heartbeat transitions into a purring vibration, indicating a relaxed state. This transition can be achieved in less than 60 s but is dependent on the device–human interaction. Further details on the logic model underlying Purrble can be found in online supplemental material 1 or see previous research.[49]

### Waitlist—control

The participants in the control group will be on a waitlist throughout the 13-week trial period. Once data have been collected at the final timepoint (week 13, T10), waitlist participants will receive a Purrble to keep. Waitlist control group was selected following discussions with Sprouting Minds members (see Patient and public involvement section).

### Participants
#### Eligibility criteria

When potential participants register their interest for the study, they will be asked demographic questions relating to the eligibility criteria, providing our information for our inclusion criteria. These are: (1) being between the ages of 16–25 years (inclusive); (2) identifying as any part of the LGBTQ+ umbrella; (3) having current experiences of self-harmful thoughts

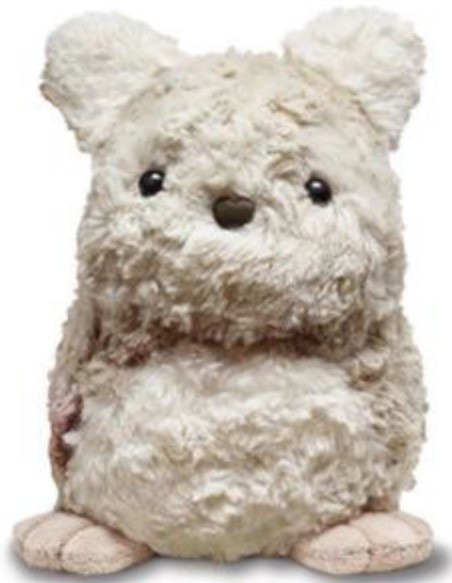

**Figure 1** Purrble—socially assistive robot.

(eg, in the last month); (4) being able to read, write and speak English; and (5) living in the UK for the duration of the study. Exclusion criteria are being outside of the age range, identifying as heterosexual cisgender, not experiencing current self-harmful thoughts, being unable to understand English and living outside the UK.

### Recruitment, randomisation and blinding

LGBTQ+ youth will be recruited to the trial by several strategies. These include: (1) approaching secondary schools and colleges, (2) social media adverts, (3) advertising through stakeholder charities and organisations (eg, Bounce Black, Harmless, King's College London newsletter) and (4) online platforms (eg, MQ Participate). Those organisations which involve gatekeepers (eg, schools, colleges, charities, organisations) will be emailed by one of the leading researchers offering an introductory meeting to discuss the outline of study, explaining safeguarding protocols and how to share this information with young people. Young LGBTQ+ people will then be able to register their interest in the study, anonymously from gatekeepers. At recruitment, participants will not be blinded to the fact that Purrble was designed to support ER. The information sheet specifies that participants will be asked about mood, self-harm and ER over the course of the study.

Once eligibility is confirmed and young LGBTQ+ people have provided written informed consent (online supplemental material 2), they will be 1:1 randomly assigned to either intervention or waitlist control group, using a computerised algorithm. A stratification procedure will be applied to balance gender identity (transgender and gender diverse youth vs cisgender) across the two arms. The researcher conducting randomisation will be blind to treatment group allocation. However, the leading researcher and other team members who will be conducting safeguarding will be aware of group allocation. Researchers collecting outcome measures will be blinded to group allocation. Participants will be informed about their assigned condition. The intervention group will receive the Purrble before T1 data are collected, with waitlist participants receiving their Purrble devices after the 3-month follow-up data collection. Participants may withdraw at any time, once they have received their Purrble device it is theirs to keep.

In the event of change of circumstance, such as a serious adverse event (eg, they are hospitalised), participants are asked to inform the research team. During the follow-up surveys, a standard operation protocol will be used where the young person will be asked "Has your situation changed at all which might impact how you'd like to engage with the study?" via email, with the understanding that the research team will have time to arrange reasonable adjustments (eg, if a young person still wants to take part but is within an inpatient service).

### Power analysis

Sample size was determined based on an a priori power analysis to detect a difference between the two arms when considering the primary outcome measure, DERS8. On the basis of a pilot with LGBTQ+ youth[47] and other Purrble studies,[50 59] we expect to see a medium effect size for this measure (d=0.4). This would indicate fewer difficulties with ER among young people who had access to Purrble.

With the anticipated medium effect size, simulations were performed involving a range of fixed and random effects. Simulations involved linearly increasing effect over the study period, and sensitivity analysis was performed over a range of scenarios considering the slope of effect change over time was either fixed or random. The simulation used a one sided t-test (=0.05) and targeted a sample size giving at least 80% power. The statistic to be compared between groups is the change in mean DERS8 Score in the 2 weeks preceding intervention (three assessments in total), to the mean DERS8 assessed at weeks 8, 9 and 10. The averaging over three assessments is intended to reduce the known variance in DERS8 when repeatedly assessed.[60] Other simulations considered only comparing the change in DERS8 from baseline (week 0) to week 10, comparing the slope of the effect, assessed using simple linear regression. All simulations suggested better than 60% power with 70 participants per arm, with the mean change in DERS8 averaged over three assessments and the comparison of slopes, suggesting >80% power. The sample size of 140 is inflated by 20% to account for dropout rate,[61] rounding up the total sample to 168.

### Outcome measures

An overview of all measures can be found in table 2 and full details can be found in online supplemental material 3. Primary and mental health measures will be asked at all timepoints, with the Purrble intervention group also receiving two additional engagement measures throughout deployment. An extended survey will replace the weekly survey at three timepoints, this will include three additional measures to be asked to all participants. All surveys will be distributed via Qualtrics using individualised links for each participant.

These additional measures were selected to explore the association between self-harm and ER (Process Model of Emotion Regulation Questionnaire—PMERQ),[38–41] perceptions of hope (State Hope Scale—SHS)[13] and loneliness (3-item University of California, Los Angeles (UCLA) Loneliness Scale),[62] to further explore the qualitative findings represented in our pilot study.[47] Our findings have previously indicated that Purrble was used to (1) refocus or distract attention during moments of distress by addressing the physical manifestation of their discomfort (PMERQ) and (2) comfort in moments of loneliness and provide self-soothing mechanisms (UCLA, SHS).[47]

Considering participant burden, young people will be informed of the time to complete each weekly survey

**Table 2** Summary of the outcome measures

| Outcome measure | Questions (n) | Type of outcome | Frequency | Scoring | Details of assessment |
|---|---|---|---|---|---|
| Primary measure | | | | | |
| Difficulties in Emotion Regulation Scale-8[60] | 8 | Primary outcome | All timepoints | 8–40 (higher=more difficulties) | Difficulties associated with response to situations eliciting negative emotions. |
| Mental health measures | | | | | |
| Self-Harm Questionnaire (SHQ) screening questions[63] | 3 | Mental health | All timepoints | Analysed separately | Frequency and risk of self-harm thoughts, suicidal ideation and behaviour. |
| SHQ additional items[63] | 22 | Covariate | Once (baseline) | Analysed separately | 4 dimensions of self-harm (NSSI, suicide attempts, suicide threats, suicide ideation). |
| Generalised Anxiety Disorder Questionnaire-7[64] | 7 | Mental health | All timepoints | 0–21 (higher=greater severity) | Presence and severity of generalised anxiety disorder. |
| Patient Health Questionnaire[65] | 9 | Mental health | All timepoints | 0–27 (higher=greater severity) | Severity of depressive symptoms. |
| Proximal and mechanistic measures | | | | | |
| Process Model of Emotion Regulation Questionnaire[66] | 9 | Mechanistic | 3 timepoints | 2 subscales Average across each subscale, higher=greater endorsement. | Attentional deployment subscales; focus on engagement and disengagement. |
| State Hope Scale[67] | 6 | Proximal | 3 timepoints | 6–48 (higher=greater state hopefulness) | Goal-directed thinking; agency and pathways. |
| UCLA Loneliness Scale for children[68] | 3 | Proximal | 3 timepoints | 3–12 (higher=more loneliness) | Subjective feelings of loneliness. |
| Engagement measures | | | | | |
| Bespoke Purrble questions[59] | 7 | Engagement | Deployment | Analysed separately | Purrble use and perceived usefulness. |
| TWente Engagement with Ehealth Technologies Scale[69] | 9 | Engagement | Deployment | 3 subscales Total score per subscale=greater engagement | Engagement with intervention device; behavioural, cognitive and affective engagements. |

Note: UCLA= University of California, Los Angeles; NSSI = Non-Suicidal Self-Injury

(15 min) and the extended survey (22 min) and will be compensated for their time.

### Post deployment interviews

We will collect semistructured interview data from LGBTQ+ young people from up to 40% of the intervention group (n=37) and approximately 20% of the control group (n=17). The interviews will be conducted following the deployment period. We will specifically aim to recruit young people who demonstrated the highest and lowest changes in the outcome data over the trial to explore and understand the potential moderators relating to the intervention and mental health across the trial period.

The semistructured interview will explore the engagement and appropriation of the Purrble device, whether LGBTQ+ young people had felt that this had helped them with their ER, mental health more broadly or self-harmful thoughts, and how Purrble may (or may not) be suitable for other audiences. We will compare these experiences between intervention and control participants, exploring other mechanisms used by LGBTQ+ youth who experience self-harm.

### Hypotheses

#### Primary hypothesis

Across the trial, we hypothesise that access to the Purrble intervention (compared with the waitlist control) will lead to a direct decrease in self-reported difficulties with ER as measured by the primary outcome (DERS8), averaged between three pre-deployment (weeks 1–3) and our final three deployment assessments (weeks 11–13).

## Secondary hypothesis

Intervention effects will be moderated by engagement with the device, measured by bespoke questions and the TWente Engagement with Ehealth Technologies Scale questionnaire. Secondary outcomes in the Purrble effectiveness trial are: self-harmful thoughts, symptoms of anxiety and symptoms of depression. These three constructs were selected as secondary outcomes based on the high prevalence of these experiences among LGBTQ+ youth[3–8] and their association with poor ER.[38–41] The three secondary hypotheses are as listed when compared with waitlist controls:

1. Engagement with the Purrble intervention will reduce the frequency of self-reported self-harmful thoughts (SHQ).
2. Engagement with the Purrble intervention will reduce the severity of self-reported anxiety symptoms (GAD-7).
3. Engagement with the Purrble intervention will reduce the severity of self-reported depression symptoms (PHQ-9).

## Additional analyses

Additional hypotheses aim to understand the impact of Purrble on relevant proximal and mechanistic outcomes. The following hypotheses will be investigated across the trial:

1. Greater within-group changes will be seen among intervention group participants, with increasing levels of endorsement for attentional deployment (PMERQ), than among those participants of the control group.
2. There will be a greater increase in state hopefulness (SHS) in the Purrble intervention group than the waitlist control.
3. Participants in the Purrble intervention group will report lower loneliness (UCLA) than those in the waitlist control group.

## Statistical analyses

Testing the hypothesis that access to the Purrble intervention will lead to a reduction in emotion dysregulation, as measured by the composite primary outcome, will be done using a one-sided t-test.

As exploratory analyses, linear mixed models will be fitted to gain insight into how emotion dysregulation is altered with access to the intervention. In particular, we will regress the weekly outcome score on an indicator for the Purrble condition, a linear time trend and an interaction between the treatment indicator and time to examine differential trends in the two groups. We will adjust for baseline covariates and include participant-level random intercepts and slopes to account for persistent baseline differences between young people as well as person-specific time trends in the outcome. While the outcome is limited to DERS8 scores ranging from 8 to 40, we will model it as continuous data.

For secondary aims, we will use analogous linear mixed models to assess the impact of Purrble on relevant outcomes (cf., hypotheses above), adding baseline DERS8 as another covariate. We will not adjust for multiple comparisons, as these are exploratory aims meant to be generate hypotheses. Similarly, we will also assess the impact of access to Purrble on changes in proximal outcomes, as well as explore whether these appear to moderate changes on primary and secondary outcomes.

## Patient and public involvement

The study design was discussed with, and approved by, Sprouting Minds members (MRC Digital Youth Young Person Advisory Group), with specific input considering the intervention arms and safeguarding procedures. These young people highlighted that 'waitlist control' conditions mimic clinical experiences of waiting for services, therefore this was considered an acceptable and realistic control. However, safeguarding procedures are included for both arms of the study to balance participant autonomy and ensure safety of research participants.

## ETHICS AND DISSEMINATION

This manuscript has been written with insights from the Spirit 2013 checklist (online supplemental material 4). The study will be conducted according to local regulations and the Declaration of Helsinki of 1975, revised in 2008. The ethical committee at King's College London, UK, approved the study (RESCM-22/23-34570). Written consent will be obtained from all participants prior to commencing their involvement in the study, with explicit understanding of the study and safeguarding procedures (see online supplemental material 5) being obtained during study briefing sessions. The trial is registered with ClinicalTrivals.gov (NCT06025942).

We aim for our findings (and any modifications to this protocol) to be disseminated across academic fields (human–computer interactions, psychology, implementation sciences), alongside showcasing the findings to LGBTQ+ youth, community groups and wider stakeholders. This will be achieved through presentations at national and international conferences, peer-reviewed journal publications, community outreach and patient and public involvement events. During dissemination, we will be liaising with youth populations to establish next steps for this research, considering additional codesign of materials to sit around/alongside Purrble.

**Author affiliations**
[1]Department of Informatics, King's College London, London, UK
[2]Institute of Mental Health, University of Nottingham, Nottingham, UK
[3]University of Glasgow, Glasgow, UK
[4]Murdoch Children's Research Institute, Parkville, Victoria, Australia
[5]University of Oxford, Oxford, UK
[6]Division of Clinical Neuroscience, University of Nottingham, Nottingham, UK
[7]Stanford University, Stanford, California, USA
[8]Division of Psychiatry, University of Nottingham, Nottingham, UK
[9]University of Nottingham, Nottingham, UK
[10]School of Psychology, University of Nottingham, Nottingham, UK

**Acknowledgements** The authors (AJW, ET, NN, AC-N, CH, PS) acknowledge the support of the UK Research and Innovation Digital Youth Programme award (MRC project reference MR/W002450/1) which is part of the AHRC/ESRC/MRC Adolescence, Mental Health and the Developing Mind programme. The authors would like to thank the PPI group of Sprouting Minds, including the two co-chairs, Sarah Doherty and Lucy-Paige Willingham, for their ongoing support and contributions to the project. RB receives salary and research support from an Australian National Health and Medical Research Council Emerging Leadership Investigator Grant (EL2; GNT2008073). The authors would also like to thank Mathijs Lucassen for his input while developing this trial.

**Collaborators** On behalf of Digital Youth: Cathy Creswell, Peter Fonagy, Louise Arseneault, Emily Lloyd, Josimar De Alcantara Mendes, Carolyn Ten Holter, Marina Jirotka, Zsofia Lazar, Praveetha Patalay, Yvonne Kelly, Aaron Kandola, Edmund Sonuga-Barke, Sonia Livingstone, Kasia Kostryka-Allchorne, Jake Bourgaize, Mariya Stoilova, Rory O'Connor, Dorothee Auer, Sieun Lee, Nitish Jawahar, Marianne Etherson, Chris Greenhalgh, Kapil Sayal, Jim Warren, Vajisha Wanniarachchi, Kevin Glover, Paul Stallard, Charlotte Hall, Mathijs Lucassen, Sally Merry, Karolina Stasiak, Camilla Babbage, Kareem Khan, Adam Parker, Joanna Lockwood, Jo Gregory, Emma Nielsen, Elvira Perez Vallejos, Rebecca Woodcock, Sarah Doherty and Lucy-Paige Willingham.

**Contributors** AJW, ET and PS conceived the study. AJW and PS designed the study, with statistical expertise from CRT and patient and public input from NN and AC-N. AJW designed the study materials, which were reviewed with feedback between NN, AC-N and SC, obtained ethical approval and drafted the first version of the protocol manuscript for publication, with input from all authors. JG, ET, PS and AJW contributed to measure selection. RB and AJW revised the submitted protocol manuscript together.

**Funding** This study is funded by the UKRI MRC Digital Youth (MR/W002450/1) and the UKRI Future Leaders Fellowship (MR/T041897/1). Funders have no involvement with the study design, data collection, management, analysis, interpretation or dissemination.

**Competing interests** PS has been involved in the development of what is now Purrble as part of his postdoctoral fellowship and serves as a paid research adviser to the Committee for Children (CfC) but has no financial stake in either CfC (Purrble brand owners) or Sproutel (company manufacturing Purrbles). Neither Committee for Children nor Sproutel had access to the data or had been part of the data collection in any way nor did they approve or see the publication before it was submitted. There were no conflicting interests among the remaining research team.

**Patient and public involvement** Patients and/or the public were involved in the design, or conduct, or reporting, or dissemination plans of this research. Refer to the Methods section for further details.

**Patient consent for publication** Not applicable.

**Provenance and peer review** Not commissioned; externally peer reviewed.

**ORCID iDs**
A Jess Williams http://orcid.org/0000-0002-3987-3824
Christopher R Tench http://orcid.org/0000-0001-9067-0494
Ellen Townsend http://orcid.org/0000-0002-4677-5958

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
