## [Reviewer comments · BMJ Open]

ARTICLE DETAILS

TITLE (PROVISIONAL)	Enhancing emotion regulation with an in-situ socially assistive robot among LGBTQ+ youth with self-harm ideation: Protocol for a randomised controlled trial
AUTHORS	Williams, A. Jess; Cleare, Seonaid; Borschmann, Rohan; Tench, Christopher; Gross, James; Hollis, Chris; Chapman-Nisar, Amelia; Naeche, Nkem; Townsend, Ellen; Slovak, Petr

VERSION 1 – REVIEW

REVIEWER	Bradwell, Hannah University of Plymouth, Faculty of Health
REVIEW RETURNED	25-Sep-2023

GENERAL COMMENTS	Thank you for inviting me to review this interesting protocol. It details plans for an important study on the impact of a social robot on the wellbeing of a minority youth group with increased risk of self-harm via enhanced emotional regulation. The planned study would have implications for youth organisations, robot developers and future research. Positive points: The use of a co-designed device is a strength, however it'll be interested to see acceptability in an older sample over this slightly longer time-frame. I can see prior work suggested continued use over 2 weeks, but this is limited time for testing novelty. The introduction and methods sections are both well written, the study is justified well based on the importance of emotional regulation for this minority group in reducing self-harm behaviours. Sample size is justified, recruitment is well explained and safeguarding considering the topic of self-harm has been considered. Minor points: The abstract could benefit from some further details. How long will they have access to the device, is it 10 weeks continued use? What is the nature of the device access, is it individual use at home, shared access at a college, group intervention? This is in the full text but could be handy in the abstract. Full stop missing on line 44 after [3-8] This sentence is difficult to understand: as it refer to 'the three' before I think what's being referred to is listed: "with mixed findings
--

	reported by the three considering the impact of the intervention on participants at this stage;” This sentence could do with a comma: “designed to provide in-the-moment support see Intervention section for further details” How will you understand the impact of dose, I can see an engagement measure, will this give accurate understanding of the number and duration of interactions? Can you clarify how the weekly surveys are collected? Will researchers ring the participants to complete surveys with them?
--	---

REVIEWER	Koh , Wei Qi The University of Queensland
REVIEW RETURNED	21-Oct-2023

GENERAL COMMENTS	Thank you for the opportunity to review this manuscript titled “Enhancing emotion regulation with an in-situ socially assistive robot among LGBTQ+ youth with self-harm ideation: Protocol for a randomised controlled trial”. Overall, this protocol is well written, well justified with methods being well detailed I have some minor comments/questions that I hope the authors can reflect on: 1) Purrble – is there a picture of Purrble that can be included for readers to have a better idea of its attributes and design? Apart from the title where the authors labelled Purrble as a socially assistive robot, no other parts of the paper has described it as such, describing it as an emotional regulation intervention tool instead. Please also state the type of robot and its features and consider staying consistent to terms used. 2) Has there been any other studies which have used socially assistive robots to support this population? There has been several studies using different social robots (e.g., telepresence, pet robots) to support the emotional wellbeing of other populations, it may be worth drawing upon this literature to consider Purrble’s similarities or differences/unique contributions, and its possible mechanism of action 3) Are there any potential ethical considerations to using Purrble?
---

VERSION 1 – AUTHOR RESPONSE

Reviewer: 1

Dr. Hannah Bradwell, University of Plymouth

Comments to the Author:

Thank you for inviting me to review this interesting protocol. It details plans for an important study on the impact of a social robot on the wellbeing of a minority youth group with increased risk of self-harm via enhanced emotional regulation. The planned study would have implications for youth organisations, robot developers and future research.

Positive points:

The use of a co-designed device is a strength, however it'll be interested to see acceptability in an older sample over this slightly longer time-frame. I can see prior work suggested continued use over 2 weeks, but this is limited time for testing novelty.

The introduction and methods sections are both well written, the study is justified well based on the importance of emotional regulation for this minority group in reducing self-harm behaviours.

Sample size is justified, recruitment is well explained and safeguarding considering the topic of self-harm has been considered.

Minor points:

The abstract could benefit from some further details. How long will they have access to the device, is it 10 weeks continued use? What is the nature of the device access, is it individual use at home, shared access at a college, group intervention? This is in the full text but could be handy in the abstract.

Engagement with the device is dependent on the participant, they are given only the instructions to "use it as much or as little as they wish", after the study the device is theirs to keep so they can continue engagement as they desire. We have added a brief statement in the abstract to convey this information:

"Intervention participants will have unlimited access to Purrble over the deployment period, which can be used as much or as little as they like. After all assessments, control participants will receive their Purrble, with all participants keeping the robot after the end of the study."

Full stop missing on line 44 after [3-8]

This is now included.

This sentence is difficult to understand: as it refer to 'the three' before I think what's being referred to is listed: "with mixed findings reported by the three considering the impact of the intervention on participants at this stage;"

We have revised this sentence to be clearer:

"These mental health interventions are typically perceived positively by LGBTQ+ youth [30, 32-34], with mixed findings reported by the three studies which considered the impact of the intervention on participants..."

This sentence could do with a comma: "designed to provide in-the-moment support see Intervention section for further details"

This is now included.

How will you understand the impact of dose, I can see an engagement measure, will this give accurate understanding of the number and duration of interactions?

We will not be measuring impact of dose directly, our weekly assessment for intervention participant includes TWEETS (engagement measure) and responses to bespoke Purrble questions which have been used in previous research (Dauden-Roquet et al., 2021; Williams et al., 2023). These items ask about i) how often Purrble was engaged with; ii) how the participant perceives this engagement; iii) and open-text responses as to situations when Purrble may or may not have been useful.

Can you clarify how the weekly surveys are collected? Will researchers ring the participants to complete surveys with them?

Thank you for pointing this out! We have now included information;

“The trial period is across 13 weeks, built of 3 pre-deployment assessments and 10 deployment assessments, using weekly, self-reported, validated surveys hosted by Qualtrics...”

“All surveys will be distributed via Qualtrics using individualised links for each participant.”

Reviewer: 2

Dr. Wei Qi Koh , The University of Queensland

Comments to the Author:

k you for the opportunity to review this manuscript titled “Enhancing emotion regulation with an in-situ socially assistive robot among LGBTQ+ youth with self-harm ideation: Protocol for a randomised controlled trial”. Overall, this protocol is well written, well justified with methods being well detailed I have some minor comments/questions that I hope the authors can reflect on:

1) Purrble – is there a picture of Purrble that can be included for readers to have a better idea of its attributes and design? Apart from the title where the authors labelled Purrble as a socially assistive robot, no other parts of the paper has described it as such, describing it as an emotional regulation intervention tool instead. Please also state the type of robot and its features and consider staying consistent to terms used.

Sorry, we’re a little confused by this comment - Figure 1 is an image of the Purrble – socially assistive robot. Purrble is described in the intervention section of the paper with additional information regarding the logic model included within supplementary materials 1. We hope this meets the requirements of the reviewer’s comments. We have revised the manuscript to be support greater consistent in terms used.

2) Has there been any other studies which have used socially assistive robots to support this population? There has been several studies using different social robots (e.g., telepresence, pet robots) to support the emotional wellbeing of other populations, it may be worth drawing upon this literature to consider Purrble’s similarities or differences/unique contributions, and its possible mechanism of action

There have not previously been studies which use socially assistive robots to support LGBTQ+ youth who have experiences of self-harm ideation. However, we have expanded our introduction to provide a review of social robot related literature.

“This is the only study to date, which has explored the impact of a socially assistive robot among LGBTQ+ youth, who are at-risk of self-harm [47].”

“Socially assistive robots (SARs) have previously been used to support children in education [51], family [49] or health settings [52-53], as well as adults with health conditions such as dementia or physical illnesses [54-55]. These studies have shown promising results in the context of motivation, skill development and enhancement, as well as supporting mental health outcomes, e.g., reducing loneliness and stress [49-55]. Similarly, students and at-risk young people have described Purrble robots are a mechanism for comfort and relieve distress [47, 50].”

3) Are there any potential ethical considerations to using Purrble?

There are several ethical considerations which we have considered, however due to word count, we have selected to discuss the challenge related to loneliness given that Purrble has been associated with comfort in the past and this relates to how we designed the study:

“However, an ethical challenge raised in SARs literature is the use of these devices as a replacement for humans, which could incur negative impacts considering social isolation [56]. Therefore, research utilising SARs should be mindful of this, considering this influence in process analysis, and have additional procedures to prevent overreliance on these devices.”